# Circulating Tumor DNA Predicts Early Recurrence Following Locoregional Therapy for Oligometastatic Colorectal Cancer

**DOI:** 10.3390/cancers16132407

**Published:** 2024-06-29

**Authors:** Conor D. J. O’Donnell, Nikolas Naleid, Teerada Siripoon, Kevin G. Zablonski, Michael H. Storandt, Jennifer E. Selfridge, Christopher L. Hallemeier, Madison L. Conces, Krishan R. Jethwa, David L. Bajor, Cornelius A. Thiels, Susanne G. Warner, Patrick P. Starlinger, Thomas D. Atwell, Jessica L. Mitchell, Amit Mahipal, Zhaohui Jin

**Affiliations:** 1Mayo Clinic School of Graduate Education, Mayo Clinic College of Medicine, Mayo Building, Rochester, MN 55905, USA; odonnell.conor@mayo.edu (C.D.J.O.);; 2Department of Medicine, University Hospitals of Cleveland, Lakeside Building, 11100 Euclid Avenue, Cleveland, OH 44016, USA; 3Division of Medical Oncology, Department of Medicine, Ramathibodi Hospital, Mahidol University, Bangkok 10400, Thailand; 4University Hospitals Seidman Cancer Center, Case Western Reserve University, Cleveland, OH 44106, USA; 5Department of Radiation Oncology, Mayo Clinic College of Medicine, Rochester, MN 55905, USA; 6Division of Hepatobiliary and Pancreatic Surgery, Department of Surgery, Mayo Clinic College of Medicine, Rochester, MN 55905, USA; 7Department of Radiology, Mayo Clinic College of Medicine, Rochester, MN 55905, USA; 8Division of Medical Oncology, Mayo Clinic College of Medicine, Rochester, MN 55905, USA

**Keywords:** colorectal cancer, circulating tumor DNA, oligometastatic disease, hepatectomy, ablation, stereotactic body radiation therapy, chemotherapy

## Abstract

**Simple Summary:**

Colorectal cancer is a major cause of cancer death, often due to metastasis. For patients with limited spread, treatments to remove all cancerous lesions can extend life or even cure the disease. However, predicting who benefits most from further treatment is challenging. This study used tumor-informed circulating tumor DNA (ctDNA) testing to detect minimal residual disease (MRD) after locoregional therapy for metastatic colorectal cancer. The results showed that positive ctDNA results after curative-intent treatment predict poor prognosis better than traditional tests. Those with negative ctDNA had over three times longer survival without recurrence compared to those with positive ctDNA. In this group of patients, the majority of whom had received prior chemotherapy, receiving more of the same chemotherapy did not seem to delay cancer recurrence. These preliminary results set the stage for future prospective trials which may examine the value of ctDNA-guided patient management for those with colorectal cancer and limited metastatic disease.

**Abstract:**

(1) Background: Local therapies offer a potentially curative approach for patients with oligometastatic colorectal cancer (CRC). An evidence-based consensus recommendation for systemic therapy following definitive locoregional therapy is lacking. Tumor-informed circulating tumor DNA (ctDNA) might provide information to help guide management in this setting. (2) Methods: A multi-institutional retrospective study was conducted, including patients with CRC that underwent curative-intent locoregional therapy to an isolated site of metastatic disease, followed by tumor-informed ctDNA assessment. The Kaplan–Meier method and log-rank tests were used to compare disease-free survival based on ctDNA results. ctDNA test performance was compared to carcinoembryonic antigen (CEA) test results using McNemar’s test. (3) Results: Our study cohort consisted of 87 patients treated with locoregional interventions who underwent ctDNA testing. The initial ctDNA test post-intervention was positive in 28 patients and negative in 59 patients. The median follow-up time was 14.0 months. Detectable ctDNA post-intervention was significantly associated with early disease recurrence, with a median disease-free survival (DFS) of 6.63 months compared to 21.30 months in ctDNA-negative patients (*p* < 0.001). ctDNA detected a numerically higher proportion of recurrences than CEA (*p* < 0.097). Post-intervention systemic therapy was not associated with improved DFS (*p* = 0.745). (4) Conclusions: ctDNA results are prognostically important in oligometastatic CRC, and further prospective studies are urgently needed to define its role in guiding clinical decisions.

## 1. Introduction

Colorectal cancer (CRC) is the second leading cause of cancer-related death worldwide, with most deaths occurring in the setting of metastatic disease [1]. In patients with oligometastatic disease, aggressive locoregional therapy to eradicate all sites of identifiable metastases can lead to prolonged survival and cure a proportion of patients. For example, the historic 5-year survival rates following complete surgical resection are between 20 and 45% in the setting of liver metastases, 25–35% for lung metastases, and up to 45% for peritoneal metastases [2,3]. Other modalities of locoregional therapy also have a role in this setting. Radiofrequency ablation has similarly demonstrated survival benefit in select cases involving the treatment of lesions smaller than 3 cm [4,5], and stereotactic body radiation therapy (SBRT) may achieve local control rates up to 92% at 2 years and also potentially improve survival [6,7]. 

Following definitive locoregional therapy to render no evidence of disease (NED), a strong evidence-based standard of care is lacking [3,8,9]. Systemic therapy in a peri-intervention or post-intervention setting has suggested disease-free survival (DFS) benefit but has failed to demonstrate overall survival (OS) improvement [10,11,12]. Thus, tools that might help identify those at higher risk of recurrence or improve patient selection for additional therapies in this setting are urgently needed. 

The first report of using circulating tumor DNA (ctDNA) in CRC to detect minimal (or molecular level) residual disease (MRD)—defined as occult tumor remaining after curative intent therapy below the resolution of standard clinical/radiological detection—involved a cohort of 18 patients mainly treated with surgical resection for isolated liver metastases [13]. Since that time, many observational studies have demonstrated the prognostic value of ctDNA testing following early-stage CRC resection [14,15,16,17]. However, there still remains a paucity of data about ctDNA dynamics and the prognostic value of these tests in the treatment of oligometastatic CRC, particularly in the setting of locoregional therapies other than surgery. 

The primary objective of the current multi-institutional study was to evaluate the prognostic value of tumor-informed ctDNA testing in routine clinical practice for those treated with locoregional therapy for CRC metastases limited to a single site of disease. 

## 2. Materials and Methods

We performed a retrospective cohort study of patients with metastatic CRC who had undergone locoregional therapy to an isolated site of metastases and subsequently had tumor-informed ctDNA testing (Signatera^TM^, Natera, Inc., San Carlos, CA, USA) [18] to assess for MRD. Patients treated at two tertiary care centers (Mayo Clinic and Seidman Cancer Center, Case Western Reserve University) and associated network sites in the states of Minnesota and Ohio were included. Eligibility criteria included ctDNA testing performed from 1 May 2019 to 1 April 2024. Institutional Review Board approvals were obtained or waived through the respective individual institutions. 

The ctDNA assay has been previously described [18]. Formalin-fixed, paraffin-embedded tumor tissue samples were used for whole-exome sequencing, identifying up to 16 tumor-specific clonal, somatic single-nucleotide variants (SNVs). Cell-free DNA was extracted from patient plasma samples. ctDNA positivity was defined as at least 2 out of 16 tumor-specific variants detected. The assay’s stated limit of detection was 0.3 tumor molecules per ml (MTM/mL), where the analytic sensitivity was reported as 95%.

The results of the ctDNA tests were recorded prior to locoregional intervention when available. Post-locoregional intervention results were characterized as assessment of MRD if they occurred more than 28 days prior to radiographic evidence of disease. Corresponding carcinoembryonic antigen (CEA) results within 4 weeks of a ctDNA result were noted. 

Patient demographics were summarized descriptively. Disease-free survival (DFS) was defined as the time from locoregional therapy for isolated metastatic disease until radiographic evidence of disease with or without biopsy confirmation. The primary objective was to determine which clinical and biochemical factors were associated with DFS post-locoregional intervention for oligometastatic disease.

The Kaplan–Meier method and log rank test were used for DFS comparisons between groups. Factors associated with DFS with *p* < 0.10 upon the use of a univariate Cox proportional hazard model were included in the multivariate analysis. Given the variability in the timing of initial MRD assessment in this retrospective cohort, the analysis was repeated, evaluating time from initial post-intervention ctDNA assessment to radiographic recurrence. A subgroup analysis of those with MRD assessment within 8-weeks post-locoregional intervention was performed, given the potential interest of this window period for clinical decision making in regard to post-intervention systemic therapy.

A descriptive analysis was performed to investigate the concordance of ctDNA and CEA results that were obtained within 4 weeks of one another. McNemar’s test for binomial proportions was used for paired data to compare recurrence detection between these two assays. Fisher’s exact test was used for comparing proportions between non-paired groups. All *p* values were 2-sided. *p* < 0.05 was considered statistically significant. Statistical analysis was carried out using BlueSky Statistics 10.3.4 and the R-4.3.1 packages survminer, survival, and coxphf.

## 3. Results

Between May 2019 and April 2024, 87 patients were included. The median follow-up from post-intervention ctDNA draw was 14.0 months. 

The median age at locoregional intervention for metastatic disease was 59 years. There was a slight male (55%) preponderance. The cohort contained 68 patients with isolated liver metastases, 8 with isolated lung metastases, and 11 with other-site (peritoneum, ovary, or lymph node) metastases. The majority of the patients (60%) had synchronous metastases at diagnosis and had received systemic therapy prior to intervention on oligometastases (88%) for a median of 15 weeks. Local therapy consisted of surgical resection (61%), ablation (7%), radiation (5%), or a combination of these treatments (27%). Other patient characteristics are described in Table 1.

The initial ctDNA test post-intervention was positive in 28 (32%) patients and negative in 59 patients (68%). Disease recurrence was noted in 58 patients (69%). Almost all the patients (96%) who had initial ctDNA positivity developed recurrence, compared to 53% of the patients who were ctDNA-negative. The median DFS was 6.63 (6.63–14.43) months for those whose first ctDNA test was positive post-locoregional intervention versus 21.30 (14.13—not reached) months for those with a negative initial ctDNA test post-intervention (Figure 1). Post-intervention ctDNA positivity (HR 2.68, 1.54–4.64, *p* < 0.001) and male sex (HR 2.04, 1.16–3.59, *p* = 0.013) were identified as prognostic factors for DFS on multivariate analysis (Figure 2, Appendix A shows multivariate and univariate analysis results). No difference in location of recurrence was identified according to post-intervention ctDNA status (Appendix A).

A total of 48 patients (55%) had ctDNA testing performed within 8 weeks post-intervention, whereas 39 patients (45%) first had ctDNA checked >8 weeks after the intervention to render no evidence of disease. To account for the variability in the time of ctDNA testing and immortal time bias, the analysis was repeated using time from post-intervention ctDNA test until recurrence (Appendix A). Post-intervention ctDNA test result was the only variable associated with DFS in both analyses.

Initial post-intervention ctDNA testing was positive in 27/58 (47%) cases that went on to develop recurrence. For patients that had already commenced post-intervention chemotherapy at the time of ctDNA testing, 3/6 (50%) recurrences were detected. This did not differ significantly from the rate of detection when MRD assessment was first performed following chemotherapy (24/52 recurrences detected, *p* = 0.390 for comparison). With repeated testing, ctDNA positivity was detected in 39/58 (67%) patients with recurrent disease at least 4 weeks prior to radiographic detection (Figure 3). 

The overall survival data were immature, as there were only seven deaths recorded at data cut-off (Appendix A). Four of the seven patients who died had a positive ctDNA post-locoregional intervention result.

### 3.1. ctDNA Dynamics

ctDNA results from prior to curative-intent locoregional intervention for oligometastatic disease were available for 33 of the 87 patients. Overall, 24 (73%) patients had a positive test, and 9 (27%) patients had a negative pre-locoregional intervention test. Pre-intervention ctDNA positivity was not a prognostic factor for post-intervention DFS (Appendix A). 

Of the 24 patients with a positive initial pre-intervention ctDNA test result, 4 patients were documented to convert to negative following pre-intervention chemotherapy. Of 20 patients whose ctDNA test remained positive pre-intervention, 10 (50%) converted to negative following locoregional therapy: 6/10 patients treated with surgery alone versus 4/10 treated with multimodality therapy; *p* = 0.656. 

In those with positive ctDNA post-intervention (n = 28), three achieved transient clearance of ctDNA; however, no patients achieved sustained clearance following cessation of systemic therapy. All but one patient (27/28) had radiographic recurrence. Of the 40 patients with anytime ctDNA positivity post-locoregional intervention, 39 patients (98%) developed recurrences. Amongst the ctDNA-positive patients, the MTM/mL value of ctDNA post-intervention was not associated with DFS (*p* = 0.729).

Figure 4 shows DFS based on the dynamic changes in ctDNA status from before to after locoregional intervention for oligometastatic disease.

The study cohort included information about subsequent treatments at time of recurrence post-locoregional intervention. We identified six patients who developed recurrent disease amenable to further locoregional therapy in the setting of ctDNA positivity during follow-up in this cohort. With further locoregional intervention, 4/6 patients subsequently converted to ctDNA-negative. Three of these patients remain ctDNA-negative and recurrence-free, whereas the fourth patient has developed another recurrence.

### 3.2. Assessment in the 8-Week Post-Intervention Window: The Role of Post-Intervention Chemotherapy

In the overall population, the administration of post-intervention chemotherapy was not prognostic of prolonged DFS (Figure 5a). No difference in baseline characteristics were noted between those who did and did not receive post-intervention systemic therapy (Appendix A). 

We performed further analysis on the 48 patients that had ctDNA testing in the 8-week period post-locoregional intervention for oligometastases, given the importance of this timepoint in clinical decision making. Chemotherapy had been received prior to intervention in 44/48 of these patients. Our exploratory analysis did not suggest a benefit of post-intervention chemotherapy in those with ctDNA positivity post-intervention (Figure 5b). 

There were 30 patients with negative ctDNA within the 8-week period post-intervention. The rate of recurrence remained high, with 18/30 patients in this group having developed recurrence. Our univariate analysis identified metachronous presentation of metastatic disease was associated with longer DFS in this group (Appendix A). Notably, post-intervention chemotherapy was not associated with improved DFS in patients with testing in this 8-week window (Figure 5c) or the overall population with negative ctDNA post-locoregional intervention (Appendix A).

### 3.3. CEA versus ctDNA

There were 192 paired ctDNA and CEA tests post-locoregional intervention that were conducted within 28 days of each other. Of these paired tests, only 122 results were concordant (64%). Among the paired samples, ctDNA was positive in 61/111 (55%) tests prior to recurrence, whereas CEA was positive in 50/111 (45.0%) tests prior to recurrence (*p* = 0.097). ctDNA was negative in 75/81 (92.6%) tests performed in surveillance amongst those without recurrence during follow-up, compared to CEA, which was negative in 61/81 (75.3%) tests (*p* = 0.011) for this group.

## 4. Discussion

In this retrospective study, ctDNA positivity was a strong predictor of early disease recurrence after locoregional therapy for oligometastatic CRC. Median DFS was more than four times longer amongst those who were ctDNA-negative post-locoregional therapy versus ctDNA-positive. This study is amongst the first and largest series to demonstrate the performance of ctDNA testing as part of standard clinical care in this stage IV CRC setting, where multiple locoregional treatment modalities are used to eradicate visible disease with curative intent. 

Our findings are generally consistent with those of more limited observational studies specifically involving ctDNA testing following surgical resection of oligometastatic disease, where ctDNA positivity has also been associated with poor prognosis [15,19,20,21,22,23].

The rates of recurrence in our study were high amongst patients with negative ctDNA post-locoregional therapy. In an 8-week window post-locoregional therapy for isolated metastatic disease, a period important for making decisions about post-intervention systemic therapy, ctDNA testing failed to detect 51% (18/35) of cases of eventual recurrence. This is in keeping with previously published data showing ctDNA testing has suboptimal sensitivity ranging from 48 to 59% during similar landmark timepoints [15,24]. The administration of post-intervention chemotherapy within 4 weeks prior to ctDNA testing did not seem to affect the performance of the assay in our limited sample, but this has been shown to affect sensitivity in previous studies [25,26]. In comparison to the traditional CEA assay, ctDNA testing still detected more recurrences overall and was negative in a higher proportion of tests in those without recurrence. Its superior performance compared to other traditional prognostic factors is in keeping with previous data [27].

This cohort predominantly (88%) included patients that had some exposure to systemic therapy prior to locoregional intervention for oligometastatic disease, as is common practice in the United States [28]. Under these circumstances, exposure to further systemic therapy post-locoregional intervention did not have a prognostic benefit in the overall cohort. 

For patients with a negative ctDNA test post-locoregional intervention, who, relatively speaking, still have better prognosis, the addition of post-intervention systemic therapy did not significantly alter recurrence risk. The non-statistically significant numerical difference in median DFS and late separation of the curves in Figure 5c seem to result from the small number of patients/events captured after 12 months of follow-up in this group. When the entire ctDNA-negative cohort was examined, there was no signal to suggest post-intervention chemotherapy was beneficial for DFS.

This finding naturally begs the question of whether post-intervention therapy could be de-escalated or omitted in patients with a ctDNA-negative assessment for MRD post-locoregional therapy for stage IV disease. The JCOG0603 trial, assessing post-operative chemotherapy following surgical resection of isolated liver metastases, has previously cast doubt about the benefit of post-intervention therapy, showing a trend towards inferior overall survival with treatment in this setting [10]. In stage II disease, where there similarly exists uncertainty about the role of post-operative systemic therapy, a ctDNA-guided strategy omitting chemotherapy in ctDNA-negative patients met its non-inferiority endpoint for DFS [29]. A similar strategy of post-intervention chemotherapy omission should be prospectively validated in the stage IV curative-intent setting. A strategy assessing de-escalation to single-agent fluoropyrimidine in patients with negative ctDNA post-hepatectomy is currently being investigated in a single-arm study (NCT05062317). 

For patients who remain ctDNA-positive post-locoregional intervention despite having received prior systemic therapy, immediate further treatment with the same systemic therapy may not significantly improve their DFS. This is supported by the fact that no patients with post-locoregional therapy ctDNA positivity in our cohort treated with post-intervention chemotherapy achieved sustained clearance of ctDNA. Almost all (29/30) of these patients developed recurrent disease, with a median DFS of less than 6 months. These results are perhaps not surprising given the lessons learned from the IDEA trial [30], where the additional benefit from 6 versus 3 months of adjuvant platinum-based chemotherapy in stage III colon cancer was, overall, fairly minimal: 0.9% absolute improvement in recurrence-free survival at 3 years from an additional 3 months of oxaliplatin-based doublet therapy [31]. In concurrence with the results of the current study, in a small subset of patients from the GALAXY trial with ctDNA positivity after oligometastatic resection, post-intervention chemotherapy did not result in prolonged DFS in those that had received pre-operative chemotherapy [15]. If MRD detection is to improve outcomes by helping select patients at highest risk of recurrence after prior systemic therapy, more effective treatments may be required, or more sensitive radiological examinations may be required to identify the occult metastatic lesions potentially amenable to further local intervention. Investigations of perioperative chemotherapy switch strategies to alternative agents are potentially warranted for patients that remain ctDNA-positive following locoregional therapy. This approach was recently validated in metastatic gastroesophageal cancer, where improved survival was achieved by switching to an alternative systemic therapy in those with stable disease after 3 months of oxaliplatin-based doublet treatment [32]. The optimal post-intervention regimen is yet to be determined. In early-stage colon cancer, treatment intensification to triplet chemotherapy with 5-fluorouracil, oxaliplatin, and irinotecan in those with ctDNA positivity following resection is currently being investigated [33] (NCT05174169). Given the effectiveness of this regimen in neoadjuvant rectal cancer [34] and in metastatic disease when combined with bevacizumab [35], it may warrant further investigation in those with positive post-locoregional intervention ctDNA in the oligometastatic setting.

This study did not formally examine patterns of imaging assessment in those with ctDNA positivity or the assay’s effect on clinical decision making. Whether more aggressive surveillance in these patients may lead to the detection of more limited disease amenable to further local intervention remains an open question. There were several cases where clinicians pursued liver MRI following the detection of ctDNA after locoregional therapy for liver metastases. Some practitioners alternatively proceeded to PET scan if CT/MRI scans were negative. Where post-intervention systemic therapy failed to lead to sustained conversion from ctDNA positivity to negativity, locoregional intervention had a significant impact on ctDNA dynamics. In all, around 50% of patients did convert to ctDNA negativity with locoregional intervention, and this was associated with improved prognosis. In our limited series, those that had converted from positive ctDNA to negative with locoregional intervention in fact had the numerically lowest recurrence rate (compared to the persistently negative and persistently positive cohorts). This may support testing both pre- and post-intervention, as ctDNA changes with locoregional intervention may be prognostically important. This finding is in contrast to those of another series involving 48 patients undergoing hepatectomy for CRC liver metastases, where the hazard ratio for recurrence-free survival was the same for negative-to-negative and positive-to-negative patients pre- and post-hepatectomy [20]. 

A strength of this study is its inclusion of various locoregional approaches, meaning it is not strictly limited to surgical resections. Our data do not suggest a significant difference in the rate of ctDNA clearance achieved between surgical resection and more multimodality approaches including ablation and radiation. Our study included patients from multiple institutions, including community sites, broadening the applicability of the results.

This pragmatic study has several limitations. It is retrospective in nature, with heterogeneity in ctDNA testing time points and frequency, precluding formal sensitivity and specificity testing. Given the variations in locoregional therapy approaches, we were not able to assess the prognostic impact of certain histopathological features on prognosis. There was also the potential for inherent selection bias in those offered ctDNA testing and post-intervention treatment, as there was variability in the insurance coverage of the test at the time these patients were treated. Finally, the duration of follow-up was limited, precluding assessment of ctDNA dynamics over time and OS comparisons.

## 5. Conclusions

The results demonstrate the prognostic value of ctDNA MRD assessment following locoregional therapy for single-site CRC metastatic disease. ctDNA positivity was strongly associated with early recurrence, generally within 7 months of locoregional therapy. ctDNA status was not a stratification factor for an associated benefit from post-intervention systemic therapy in our cohort of heavily pre-treated patients. Perioperative ctDNA dynamics are likely to continue to gain more widespread use in clinical practice, but prospective randomized data should be sought to better evaluate their utility in clinical decision making.

## Figures and Tables

**Figure 1 cancers-16-02407-f001:**
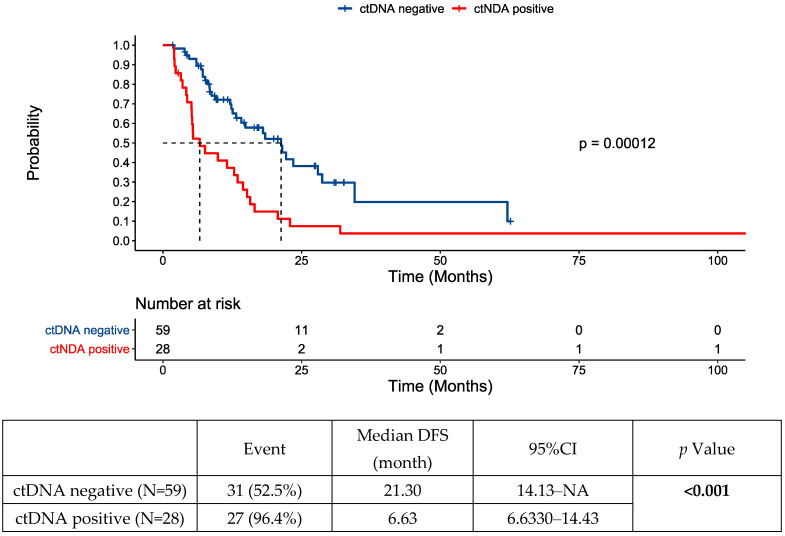
Disease-free survival (DFS) following locoregional therapy for oligometastatic disease by post-intervention circulating tumor DNA (ctDNA) status. NA—not assessed/not reached.

**Figure 2 cancers-16-02407-f002:**
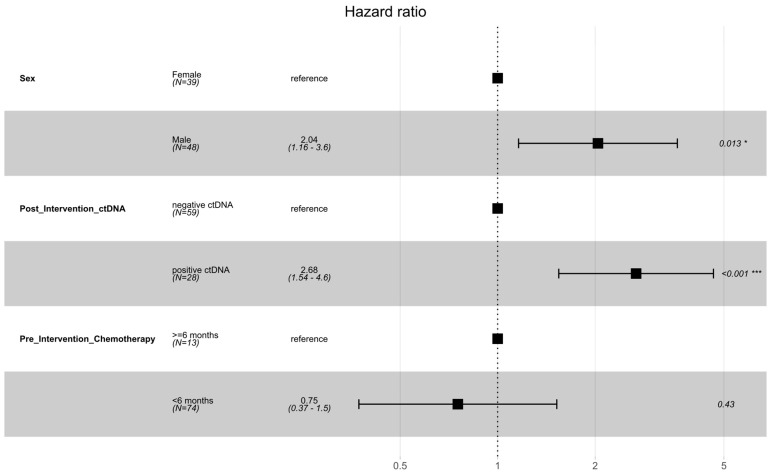
Forest plot of multivariable analysis of prognostic factors for DFS following locoregional therapy for oligometastatic disease. Significance codes: * *p* < 0.05; *** *p* < 0.001.

**Figure 3 cancers-16-02407-f003:**
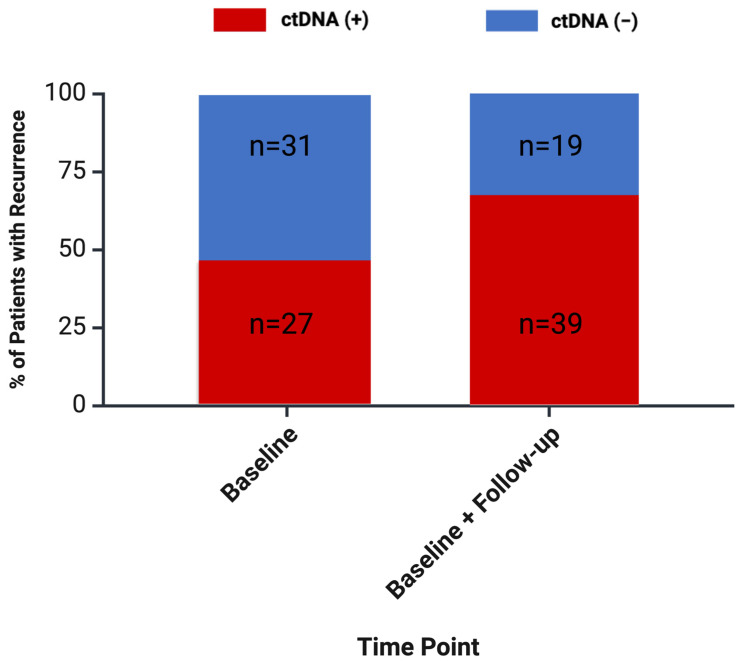
Percentage of recurrences detected with ctDNA testing for minimal residual disease (MRD). Assessment of MRD was defined as MRD occurring more than 28 days prior to radiographic evidence of disease.

**Figure 4 cancers-16-02407-f004:**
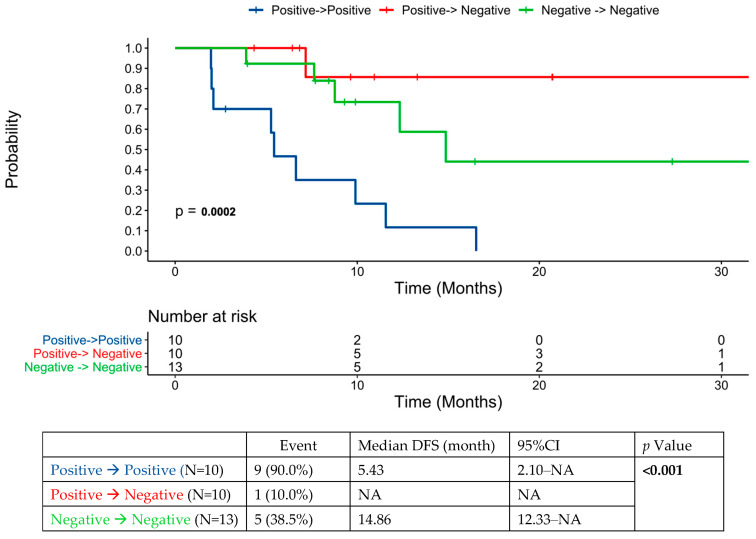
ctDNA dynamics pre- and post-locoregional therapy for oligometastatic disease in the cohort of patients that had ctDNA testing performed prior to locoregional therapy.

**Figure 5 cancers-16-02407-f005:**
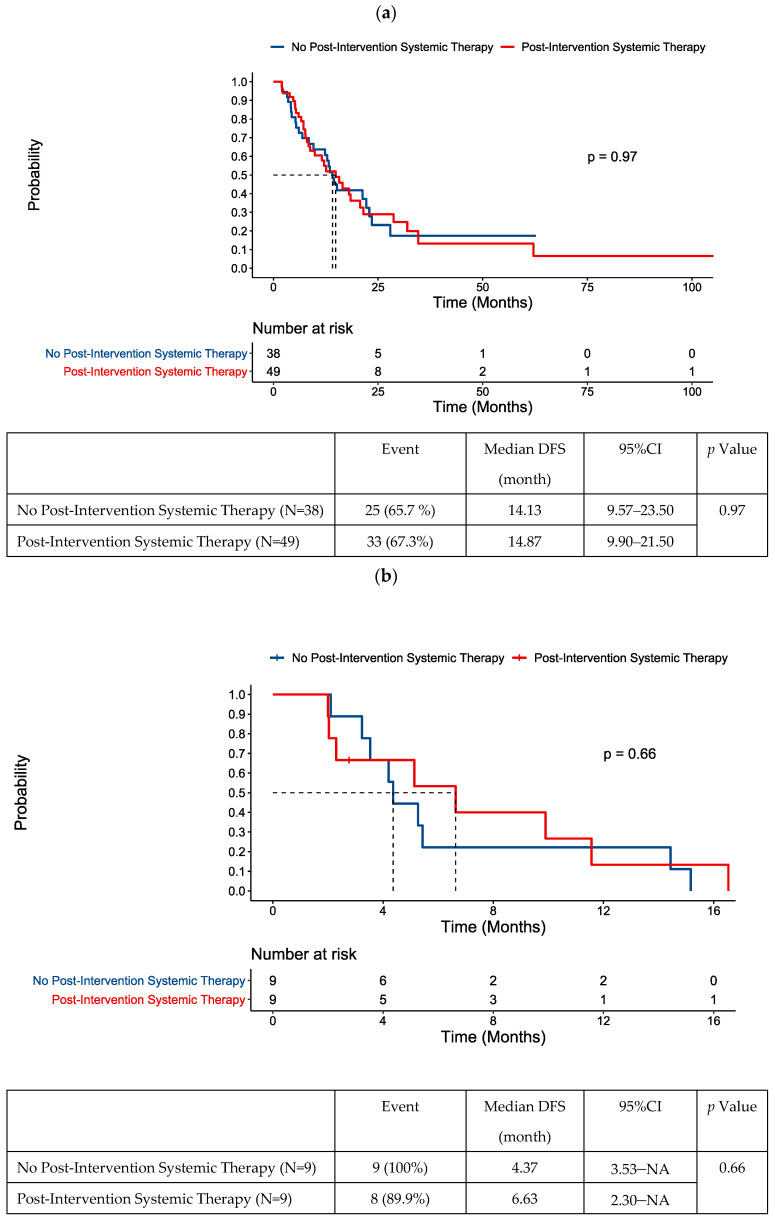
DFS following locoregional therapy for oligometastatic disease according to post-intervention systemic therapy status in (**a**) the whole cohort, (**b**) patients with positive ctDNA within an 8-week window post-intervention, and (**c**) patients with negative ctDNA within an 8-week window post-intervention.

**Table 1 cancers-16-02407-t001:** Baseline characteristics of patients with colorectal cancer treated with locoregional therapy to an isolated site of metastatic disease followed by circulating tumor DNA assessment.

Characteristics	All Patients N = 87 (%)
Median age (IQR), years	59 (32–86)
-<65	62 (71)
-≥65	25 (29)
Sex	
-Male	48 (55)
-Female	39 (45)
Ethnicity	
-White	70 (81)
-Black	14 (16)
-Other	3 (3)
Molecular profile of tumor	
-KRAS/NRAS mutation	
• Positive	36 (46)
• Negative	43 (54)
• Unknown	8
-BRAF mutation	
• Positive	1 (1)
• Negative	75 (99)
• Unknown	11
-MSI/MMR	
• dMMR	2 (2)
• pMMR	85 (98)
Initial stage at diagnosis	
-Stage I	4 (5)
-Stage II	8 (9)
-Stage III	23 (26)
-Stage IV	52 (60)
Location of primary tumor	
-Right-side	23 (26)
-Left side	64 (74)
Pattern of metastatic disease	
-Synchronous metastases	52 (60)
-Metachronous metastases	35 (40)
Location of isolated metastatic disease	
-Liver	68 (78)
-Lung	8 (9)
-Others	11 (13)
Median amount of pre-intervention systemic therapy (IQR), weeks	15 (0–58)
Pre-intervention systemic therapy regimen	
-None	11 (12)
-Single agent chemotherapy	5 (6)
-Doublet chemotherapy	58 (65)
-Triplet chemotherapy	12 (14)
-Other	3 (3)
ctDNA pre-intervention	
-Not tested	54 (62)
-Tested	33 (38)
• Positive	24 (73)
• Negative	9 (27)
Type of local therapy to metastatic disease	
-Resection	53 (61)
-Ablation	6 (7)
-Stereotactic body radiation therapy	4 (5)
-Multimodality	24 (27)
ctDNA post-intervention	
-Positive	28 (32)
-Negative	59 (68)
Median time from definitive therapy to ctDNA test (IQR), weeks	6.1 (0.4–436.9)
-≤8 weeks	48 (55)
->8 weeks	39 (45)
Post-intervention systemic therapy	
-Yes	49 (56)
-No	38 (44)

## Data Availability

The datasets cannot be made available publicly due to the respective institutional policies.

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
