# Peer review of "Circulating Tumor DNA Predicts Early Recurrence Following Locoregional Therapy for Oligometastatic Colorectal Cancer"

_cancers, 2024, doi:10.3390/cancers16132407_

Round 1

Reviewer 1 Report

Comments and Suggestions for Authors

Dear Author

·        The “ctDNA positive” should be defined. What is cut-off?

·        Both total extraction yield and the detection of mutant alleles

·         The ethics should include both committee name and ethical code the sentence “Ethical review and approval were waived for this study 354 due to its retrospective nature and low perceived risk of harm.” Is not enough

·        I don’t have an instrument to check similarity/plagiarism.

Comments on the Quality of English Language

·        The “ctDNA positive” should be defined. What is cut-off?

·        Both total extraction yield and the detection of mutant alleles

·         The ethics should include both committee name and ethical code the sentence “Ethical review and approval were waived for this study 354 due to its retrospective nature and low perceived risk of harm.” Is not enough

·        I don’t have an instrument to check similarity/plagiarism.

Author Response

Thank you very much for your feedback and constructive comments. Please see revised manuscript with tracked changes.

Comment 1: The “ctDNA positive” should be defined. What is cut-off? -Both total extraction yield and the detection of mutant alleles.

Response 1: The limit of detection and a more thorough description of the assays used is added to the methods section, along with the initial reference manuscript for this assay in colorectal cancer. The "total extraction yield" is not available.

Comment 2: "The ethics should include both committee name and ethical code the sentence “Ethical review and approval were waived for this study 354 due to its retrospective nature and low perceived risk of harm.” Is not enough".

Response: The ethics statement at the end of the manuscript is changed to add the committee names and ethical code.

Reviewer 2 Report

Comments and Suggestions for Authors

thank you for allowing me to review this original retrospective study evaluating the impact of tumor DNA circulating after management of a single metastatic recurrence of colorectal cancer. the manuscript and well written, the main and secondary objectives clearly expressed. although the size of the study is small, some interesting results are suggested and the discussion evokes the limitations of the study. 

however, I have a few remarks and comments. 

Is it lawful to have included both synchronous and metastatic metastases whose natural histories and evolutions are known to be radically different?

44% of patients treated for metastatic recurrence did not receive adjuvant chemotherapy. What were the reasons? 

what are the raiosn of the lack of preoperative assay of tumor DNA crculant in 62% of patients included?  

In the end, it seems that the post-operative detection of circulating tumor DNA is a factor not of tumor recurrence but of survival without recurrence lower. what therapeutic positions could be considered? change of chemotherapy? intensification? closer monitoring? it would be interesting to discuss them. 

Author Response

Comment 1: "Is it lawful to have included both synchronous and metastatic metastases whose natural histories and evolutions are known to be radically different?"

Response 1:  We agree that synchronous and metastatic metastases may have different natural histories. ctDNA positivity assessed in this study was a poor prognostic factor regardless of metachronous or synchronous disease. The main randomized trials that assess systemic therapy in the setting of isolated metastases (Nordlinger 2013 and Kanemitsu 2021) have included both synchronous and metachronous patient presentations. Practice guidelines also do not give clear distinctions for management of these patients based on this status.

Comment 2: "44% of patients treated for metastatic recurrence did not receive adjuvant chemotherapy. What were the reasons?"

Response 2: No patient factors were identified that correlated with decision to receive pseudoadjuvant therapy (supplemental table S4 is now added, line 206). The reasons may be difference in physician practice given the equipoise surrounding the benefit of treatment in this setting. It is also possible some factor not captured, such as post-intervention patient fitness, is responsible.

Comment 3: "what are the raiosn of the lack of preoperative assay of tumor DNA crculant in 62% of patients included? "

Response 3: At the time of this study, there was no particularly compelling data that ctDNA results in the setting of visible metastatic disease (pre-intervention) had prognostic or predictive clinical value. Thus, there is likely a difference in intra-/inter-physician practice in this retrospective study without a standardized collection protocol.

Comment 4:  "what therapeutic positions could be considered? change of chemotherapy? intensification? closer monitoring? it would be interesting to discuss them."

Response 4: Please see line 309-315 and 320-322 for some added discussion about potential interventions.